# Mapping Winter Wheat with Optical and SAR Images Based on Google Earth Engine in Henan Province, China

Changchun Li [1,†], Weinan Chen [1,*,†], Yilin Wang [1], Yu Wang [1], Chunyan Ma [1], Yacong Li [1], Jingbo Li [1,2] and Weiguang Zhai [1]

1 School of Surveying and Land Information Engineering, Henan Polytechnic University, Jiaozuo 454000, China; lichangchun610@126.com (C.L.); 211904010019@home.hpu.edu.cn (Y.W.); wangyuchn@whu.edu.cn (Y.W.); mayan@hpu.edu.cn (C.M.); 211904020026@home.hpu.edu.cn (Y.L.); lijingbo1024@163.com (J.L.); 212104020047@home.hpu.edu.cn (W.Z.)
2 Beijing Research Center for Information Technology in Agriculture, Beijing Academy of Agriculture and Forestry Sciences, Beijing 100097, China
* Correspondence: 211904020046@home.hpu.edu.cn; Tel.: +86-13276900532
† These authors contributed equally to this work.

**Abstract:** The timely and accurate acquisition of winter wheat acreage is crucial for food security. This study investigated the feasibility of extracting the spatial distribution map of winter wheat in Henan Province by using synthetic aperture radar (SAR, Sentinel-1A) and optical (Sentinel-2) images. Firstly, the SAR images were aggregated based on the growth period of winter wheat, and the optical images were aggregated based on the moderate resolution imaging spectroradiometer normalized difference vegetation index (MODIS-NDVI) curve. Then, five spectral features, two polarization features, and four texture features were selected as feature variables. Finally, the Google Earth Engine (GEE) cloud platform was employed to extract winter wheat acreage through the random forest (RF) algorithm. The results show that: (1) aggregated images based on the growth period of winter wheat and sensor characteristics can improve the mapping accuracy and efficiency; (2) the extraction accuracy of using only SAR images was improved with the accumulation of growth period. The extraction accuracy of using the SAR images in the full growth period reached 80.1%; and (3) the identification effect of integrated images was relatively good, which makes up for the shortcomings of SAR and optical images and improves the extraction accuracy of winter wheat.

**Keywords:** winter wheat; Sentinel; Google Earth Engine; image aggregation; integrated image; random forest

## 1. Introduction

Agricultural production is the basis of a country's socio-economic development and is the key to land resource management and food security [1]. Winter wheat is the most widely planted crop in the world, and its planting area and production are important for a country to make economic development plans, regulate crop planting structure and ensure social stability [2,3]. The timely and accurate acquisition of winter wheat acreage is crucial to the formulation of agricultural policies. At present, a variety of land cover products with different resolutions have been developed, such as the WorldCover2020 product with 10 m resolution produced by the European Space Agency (ESA), the MCD12Q1 product [4] with 500 m resolution produced by Boston University, and the GlobeLand30 product [5] with 30 m resolution produced by the National Geomatics Center of China (NGCC). These remote sensing products provide strong data support for studying the spatial distribution of farmland, but few can provide information of specific crops.

The traditional acquisition and updating of crop acreage and distribution information generally require managers to conduct field visits or consult local agricultural statistical reports [6]. However, this process is tedious and consumes a lot of human and material

resources, and it can produce some unpredictable errors. With the development of remote sensing technology, it is possible to obtain the crop acreage and distribution of crops quickly and accurately. MODIS images are widely used in crop identification due to their high revisit cycle and easy acquisition. However, due to the low spatial resolution, a large number of mixed pixels is produced, which affects the extraction of the planting area and the development of high-precision crop spatial distribution products [7]. The use of remote sensing images with medium-high spatial resolution as a data source is the key to solving this problem [8,9]. This kind of image has been widely used in the extraction of crop planting structures and the determination of the optimal time window for crop recognition. Limited by the computing power, the existing literature usually takes small-scale areas as the study object to reduce the time spent in image processing [10–12].

Google Earth Engine (GEE) is a non-profit cloud computing platform for geographic spatial analysis [13]. It has been widely used in large-scale remote sensing studies, including forest monitoring [14], crop yield estimation [15], and crop mapping [16]. The multi-platform image dataset in GEE and its derivatives provide a stable data source for crop extraction using multi-source remote sensing images. The powerful data parallel computing capability of the GEE platform provides a technical guarantee for the processing of remote sensing big data.

Optical remote sensing image is a key data source used in crop extraction studies. The use of this type of image to complete crop identification and area monitoring has been quite mature [17]. A plethora of crop mapping work has been conducted in Henan Province based on multi-source optical remote sensing images [18,19]. However, the extraction of crop information is limited due to the influence of cloud and rain weather on optical remote sensing images [20]. Compared with optical technology, radar technology has the advantage of acquiring images all day and in all weather conditions. The penetration of radar satellites can acquire the surface information of vegetation. Also, it reflects the structural changes of plant stems and leaves under various weather conditions. The previous studies show the practicality and feasibility of using SAR images for crop extraction [21,22].

Optical and SAR images have their own advantages. The effective integration of them can improve the extraction accuracy of crop acreage. Some scholars have successfully used integrate images to extract crop planting information [23,24]. The classification accuracy was limited by the number of bands of SAR images and the speckle noise in the images, and crops were extracted using only SAR images [25,26]. After integrating the texture features of Sentinel-1 data and Landsat-8 data, it was found that the integrated data could increase the extraction accuracy of winter wheat [27]. Therefore, Sentinel-1A and Sentinel-2 images with a spatial resolution of 10 m were selected as the data source for the experiment, and the texture features in the polarization feature band were calculated.

Since the resolution of Sentinel-2 images is affected by clouds and shadows, the use of cloud removal, curve smoothing, and linear interpolation methods can reduce the noise effect of clouds to some extent, but it cannot fundamentally eliminate local noise [7,17]. Thus, it is a great challenge to construct optical images for the whole fertility period. According to the characteristics of optical images and SAR images, this paper proposes image integration methods that solve the problem of missing optical images and make full use of SAR images. The proposed method provides a new idea for the extraction of winter wheat.

This study mainly explores the potential of using Sentinel-1A and Sentinel-2 images to extract winter wheat acreage and draw an accurate spatial distribution map of winter wheat. The main objectives of this study are: (1) to propose image aggregated schemes for different sensors based on the full use of sentinel images; (2) to determine whether Sentinel-1A images can be used to effectively distinguish winter wheat from other types of ground objects in the whole growth period; and (3) to determine whether the integration of Sentinel-1A images and before-wintering and after-wintering Sentinel-2 images can extract winter wheat more accurately.

## 2. Study Area and Datasets

### 2.1. Study Area

Henan Province is located in the middle and lower reaches of the Yellow River in the southern part of the North China Plain (between 31°23′–36°22′ N and 110°21′–116°39′ E) with a total area of 167,000 km² (Figure 1). The overall terrain of Henan Province is high in the west and low in the east. There are tall and undulating mountains in the west and vast plains in the east of Henan. The province is dominated by plains, with the Yellow Huaihai Alluvial Plain in the middle and east, and the Nanyang Basin in the southwest. Henan Province has a subtropical-warm temperate, humid-semi-humid monsoon climate. The annual average temperature from north to south is 10.5–16.7 °C; the frost-free period is 201–285 days; the annual average sunshine is 1285–2292 dh, and the annual average precipitation ranges from 407.7–1295.8 mm [28]. The geographical location and climate of Henan Province provide suitable growing conditions for winter wheat. According to the agricultural information released by the Department of Agriculture and Rural Affairs of Henan Province, and combined with the field survey data, the final phenological period of winter wheat in Henan Province from 2018 to 2019 was finally determined [29].

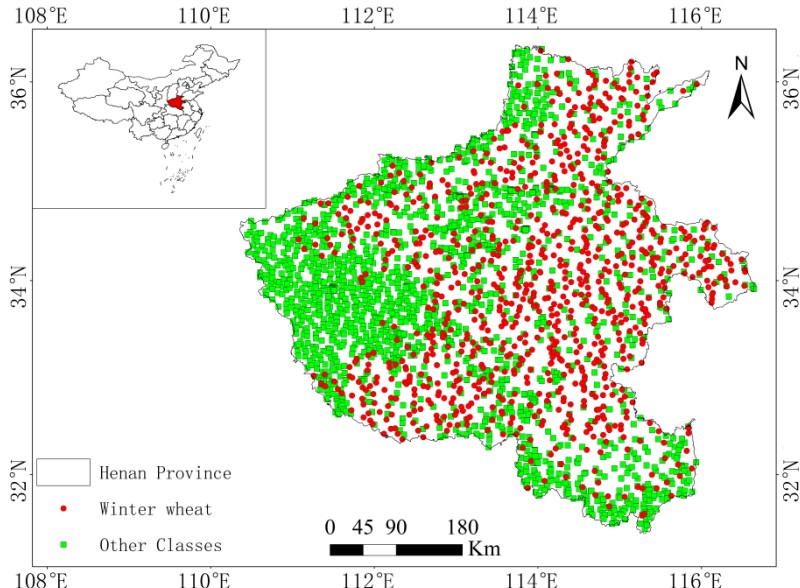

**Figure 1.** The study area in Henan Province, China. The spatial locations of the training and validation samples are shown in the figure.

### 2.2. Datasets and Preprocessing

#### 2.2.1. Sentinel Data

Sentinel-1 and Sentinel-2 satellites are both earth observation satellites in the Copernicus program of the ESA [30]. Sentinel-1 consists of 2 polar-orbiting satellites, both carrying C-band (5.4 GHz) sensors with a single satellite revisit period of 12 days. In this study, the Sentinel-1A ground range detected (GRD) product in the IW mode was taken as the data source. It has two polarization imaging modes: cross-polarization VH and co-polarization VV (Table 1). The Sentinel-1A GRD images in GEE were processed by the Sentinel-1 toolbox, including thermal noise removal, radiometric correction, terrain correction using the digital elevation model (DEM), and conversion of the backscattering coefficient to decibels (dB). After the successful launch of the Sentinel-2B, the temporal resolution of Sentinel-2 images was increased to five days, and both satellites carry a multispectral instrument (MSI) with an orbital width of 290 km [31]. The MSI images acquired by this satellite possess 13 bands covering visible, near-infrared, and short-wave infrared with spatial resolutions of 10 m, 20 m, and 60 m, respectively (Table 1). This study used MSI L1C images as the data source, which is the atmospheric apparent reflectance product after orthophoto correction and

geometric fine correction. The images with a cloud score below 15% were retained, and the cloudy pixels were removed by using the QA60 band [17,18]. This cloud removal method directly performs bit-by-bit operation on the QA60 quality band to filter pixel values. Meanwhile, it masks pixels such as cloud, cirrus cloud, rain, and snow, and finally achieves the effect of cloud removal. In total, 372 Sentinel-1A GRD images from 1 October 2018 to 15 June 2019, 522 Sentinel-2 MSI L1C images from 1 October 2018 to 30 November 2018 and 1 February 2019 to 20 April 2019 were used in this study (Figure 2).

**Table 1.** Characteristics of the satellite data used in this study.

|  | Sensor | Band | Wavelength | Resolution |
|---|---|---|---|---|
| Sentinel data | Sentinel-1A GRD | VV |  | 10 m |
|  |  | VH |  | 10 m |
|  | Sentinel-2 MSI | Blue | 490 nm | 10 m |
|  |  | Green | 560 nm | 10 m |
|  |  | Red | 665 nm | 10 m |
|  |  | Near-infrared | 842 nm | 10 m |

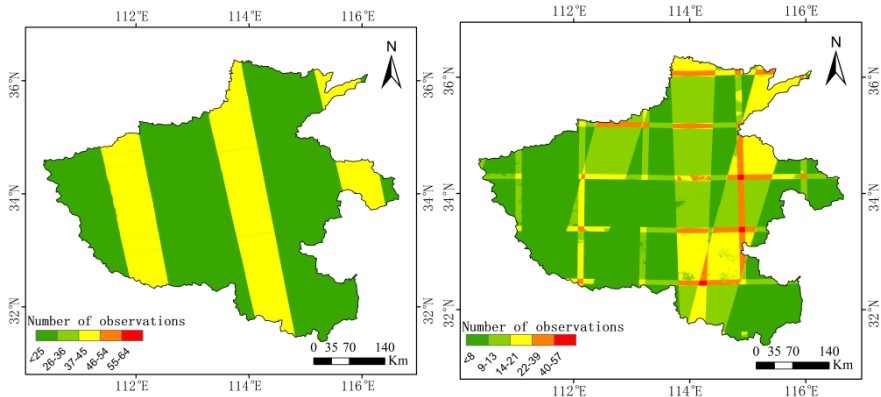

**Figure 2.** The number of good-quality observations of Sentinel-1A (**left**) and Sentinel-2 (**right**).

### 2.2.2. Sample Data

The construction and validation of the classification model require reliable ground sample data. This study used a semi-automated hierarchical sample data acquisition method to obtain ground sample points. First, the official statistical data of Henan Province was used to analyze the planting structure of winter crops in various cities, which provides a priori knowledge for the correction of sample points. Then, the national land use data provided by the Chinese Academy of Sciences in 2018 (CNLUCC2018) [32] was reclassified, and the reclassified images were used to generate sample points. Finally, the sample points were corrected by visual interpretation based on the Google Earth (GE) platform. According to the study area profile and the study purpose, the sample data were classified into five land-cover types, namely, winter wheat, vegetation, water, buildings, and others. To avoid overfitting, the sample data were randomly divided into training and validation datasets at a ratio of 7:3. The number of pixels per class is shown in Table 2.

### 2.2.3. MODIS Data

The MODIS and NDVI vegetation index products acquired by Terra and Aqua satellites were used in this study [33]. The spatial resolution of the two products is 250 m, and the temporal resolution is 16 days. Since the acquisition of the images did not overlap in time, the two products were combined into NDVI time-series data with a time series of 8 days. This time-series data was used to determine the integration time window for the Sentinel-2 images. A more accurate image aggregation time window can be obtained by removing the influence of cloud and atmosphere on the NDVI time series curve. In this

study, Savitzky-Golay filtering (S-G filtering) was adopted to smooth the NDVI time-series data to eliminate irregular fluctuations and acquire NDVI curves that are more consistent with the growth of winter wheat [7].

**Table 2.** The Sample data selected in this study.

| Land-Cover Types | Description | Samples |
|---|---|---|
| Winter wheat | Winter wheat during the observation period | 910 |
| Vegetation | Other crops, evergreen forest, deciduous forest, etc. | 900 |
| Water | Rivers, reservoirs, and lakes, etc. | 210 |
| Building | Residential land, roads, etc. | 480 |
| Other | Wasteland, unused land, etc. | 290 |

## 3. Methodology

The workflow of the study is shown in Figure 3. First, the Sentinel-1A images were aggregated according to the growth period of winter wheat (Section 3.1.1), and the Sentinel-2 images were aggregated according to the image aggregation time window selected by MODIS-NDVI curve (Section 3.1.2). In this way, a total of six Sentinel-1A images and two Sentinel-2 images were generated. Then, two polarization features, five spectral features, and four texture features were selected as features variables (Section 3.2). Subsequently, 12 experimental schemes were constructed (Section 3.3), and winter wheat and other land-cover types were classified with the RF algorithm (Section 3.4). Finally, the accuracy of each experimental scheme was evaluated (Section 3.5), and an accurate spatial distribution map of winter wheat in Henan Province was drawn based on the integrated image (scheme 12). Also, the feature variable importance was evaluated based on the classification results of the SAR images in the full growth period (scheme 6) and the aggregated image (scheme 12, Section 3.6).

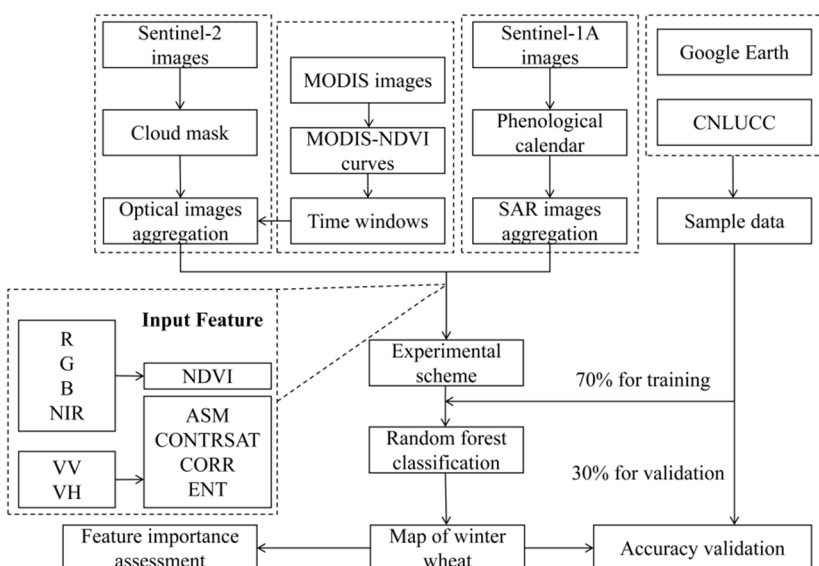

**Figure 3.** Flow chart of the overall workflow.

### 3.1. Image Aggregation Scheme

Due to the influence of the satellite revisit cycle and weather conditions, the study area may be covered by multiple images or not covered in a period of time. As for the former case, the median value of the covering images was calculated during the observation time period; as for the latter case, the image filtering time window was increased to complete image aggregation [34]. Images aggregation can take full use of remote sensing images and enhance image information. According to the study objectives and the characteristics of

Sentinel-1A and Sentinel-2 images, two different image aggregation methods are proposed in the paper.

### 3.1.1. MODIS-NDVI Curves and Aggregation of Sentinel-2 Images

Under the influence of cloud, rain, light, and other factors, the quality of Sentinel-2 images cannot be guaranteed. Thus, it is impossible to accurately acquire all remote sensing images in the observation time period. The images can be repaired by contemporaneous image masks or linear interpolation, but the final results cannot accurately reflect the changes of the images within the time period [35]. To ensure the authenticity and reliability of the data, the image aggregation time window is determined according to the MODIS-NDVI time series to complete image aggregation.

The timing of winter wheat phenology changes with the increase of latitude, and the change is obvious for the increase of two degrees. To determine the accurate Sentinel-2 image aggregation time window, Henan Province was divided into three regions (31°23′–33° N, 33°–34°42′ N, 34°42′–36°22′ N) according to latitude [36]. Then, 33 sample points were taken from each region to draw the average curve of NDVI for the winter wheat in each region (Figure 4, left). The aggregated images were selected in the period when the NDVI values of three regions increases. The aggregation time windows of Sentinel-2 images and the number of images are shown in Table 3. The images from 23 October 2018 to 30 November 2018 and the images from 10 February 2019 to 3 April 2019 were selected to aggregate Sentinel-2 median images, and the images were defined as before-wintering and after-wintering images, respectively.

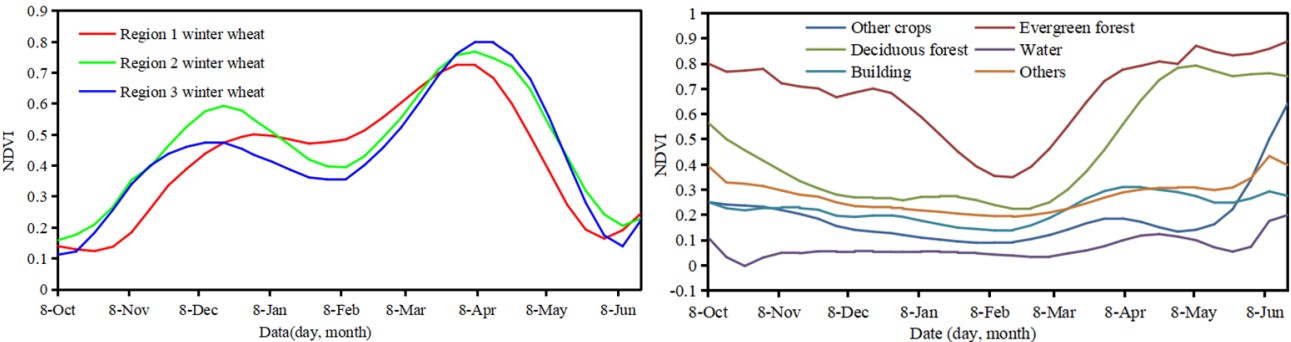

**Figure 4.** MODIS average NDVI time series curves for winter wheat (**left**) and other land cover types (**right**) in the study area from 8 October 2018 to 18 June 2019.

**Table 3.** The aggregation time window of Sentinel-1A and Sentinel-2 images. The number of images available for each aggregation time window in Henan Province.

| Data | Crop Development Period | Image Acquisition Dates | Number of Images |
|---|---|---|---|
| Sentinel-1A | Sowing | 1 October–31 October | 32 |
| | Seedling and tillering | 1 November–31 November | 40 |
| | tillering and over-wintering | 1 December–31 January | 80 |
| | over-wintering and reviving | 1 February–31 March | 77 |
| | jointing and heading | 1 April–30 April | 40 |
| | flowering and maturing | 1 May–15 June | 58 |
| Sentinel-2 | before-wintering | 1 October–30 November | 335 |
| | after-wintering | 1 February–20 April | 248 |

Before the overwintering period, with the growth of winter wheat, the influence of soil background on the NDVI curve gradually weakened, and NDVI increased and reached the first peak. After the overwintering period, the chlorophyll content in the winter wheat decreased and basically stopped growing. Thus, NDVI decreased and reached the valley peak. After reviving and before heading of the winter wheat, the NDVI curve of winter

wheat increased until reaching the second peak. After that, the winter wheat entered flowering and maturing, and the NDVI value gradually decreased to the minimum.

In this study, 400 sample points of various ground features (100 for each land-cover type, except winter wheat) were selected to acquire the pixel values of various sample points and calculate the average value of NDVI. The NDVI curves of water, buildings, and other land-cover types fluctuate little with time, which makes it easy to distinguish winter wheat curves. Affected by phenology and weather conditions, winter wheat, and vegetation have their unique NDVI curve characteristics. The NDVI curves of deciduous forest, evergreen forest, and other crops are presented respectively, although they are all divided into vegetation. Affected by phenology and weather, various types of vegetation show their unique NDVI curve characteristics.

### 3.1.2. Growth Period of Winter Wheat and Aggregation of Sentinel-1A Images

Synthetic aperture radar is not affected by cloud and rain, and it can acquire images all day and all night [37]. This study used all winter wheat observation images to give full play to the advantages of SAR images. Taking the growth period of Winter Wheat in Henan Province as the aggregation time window, the median value was calculated pixel by pixel. The SAR images were aggregated for six different growth periods, namely (1) sowing, (2) seedling and tillering, (3) tillering and over-wintering, (4) over-wintering and reviving, (5) jointing and heading, and (6) flowering and maturing. The aggregation time windows of Sentinel-1A images and the number of images are shown in Table 3.

The mean VV and VH curves of various land-cover types are illustrated in Figure 5 to demonstrate the changes of features over time. The polarization feature curve of winter wheat is quite different from that of other land-cover types. The backscattering coefficient values of the two polarization features showed an "increase-decrease-increase" trend, and the change of VV is particularly obvious.

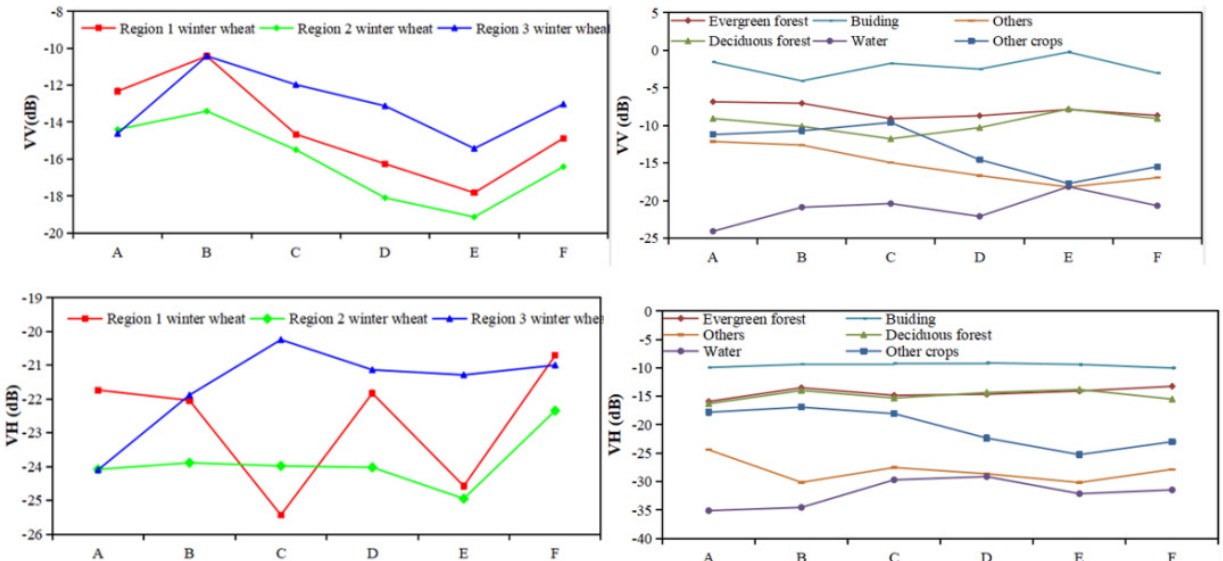

**Figure 5.** Time series of Sentinel-1A curves for each land cover class. A. B, C, D, E, and F represent the SAR images of sowing, seedling and tillering, tillering and over-wintering, over-wintering and reviving, jointing and heading, and flowering and maturing winter wheat, respectively.

### 3.2. Calculation of Feature Variables

After comprehensively considering the ecological environment of the study area, the structure of various ground features, the phenological features of winter wheat, as well as the significance of various feature variables, this study selected five spectral features, two polarization features, and four texture features as feature variables. Specifically, the R, G, B, and NIR bands in Sentinel-2 images are four common spectral bands for agricultural

monitoring with a spatial resolution of 10 m. NDVI is sensitive to chlorophyll absorption. It is the best indicator for detecting vegetation growth status and is widely used for vegetation identification [33]. Polarization features VV and VH can reflect the change of water content in plant canopy with the plant growth cycle [38].

Texture embodies the surface or structural properties of the image and can be used as a feature variable to improve the accuracy of vegetation classification [39–41]. This study used a gray level co-occurrence matrix to generate texture features in VV and VH bands. The experiments on sliding window size ($3 \times 3, 5 \times 5, 7 \times 7$) indicated that the sliding window of $3 \times 3$ was appropriate. A large number of texture features can cause data redundancy. On the premise of retaining the maximum amount of information and not exceeding the calculation limit of the GEE platform, this study selected angular second moment (ASM), contrast (CONTRAST), correlation (CORR), and entropy (ENT) as characteristic variables to improve the classification accuracy [42]. Specifically, ASM reflects the degree of coarseness and the uniformity of grayscale distribution of the texture; CONTRAST reflects the depth of texture grooves and the clarity of the image; CORR reflects the consistency of the texture in the local area; and ENT reflects the complexity of the texture.

### 3.3. Experimental Design

According to the study purposes, 12 experimental schemes (Figure 6) were designed. To explore the feasibility of mapping the spatial distribution of winter wheat, the SAR images were aggregated based on the growth period of winter wheat (i.e., from sowing to flowering and maturing). The extraction accuracy of the experimental schemes was acquired separately (schemes 1–12). To investigate the potential of using aggregated images for winter wheat acreage extraction, three experimental schemes based on aggregated integration images were designed (schemes 8, 10, and 12). The polarization and texture features of SAR images and the spectral features of optical images were extracted, and these three types of bands were taken as feature variables to extract the planting area of winter wheat. Meanwhile, a control group experiment was also designed to extract winter wheat acreage using only optical images (schemes 7, 9, and 11).

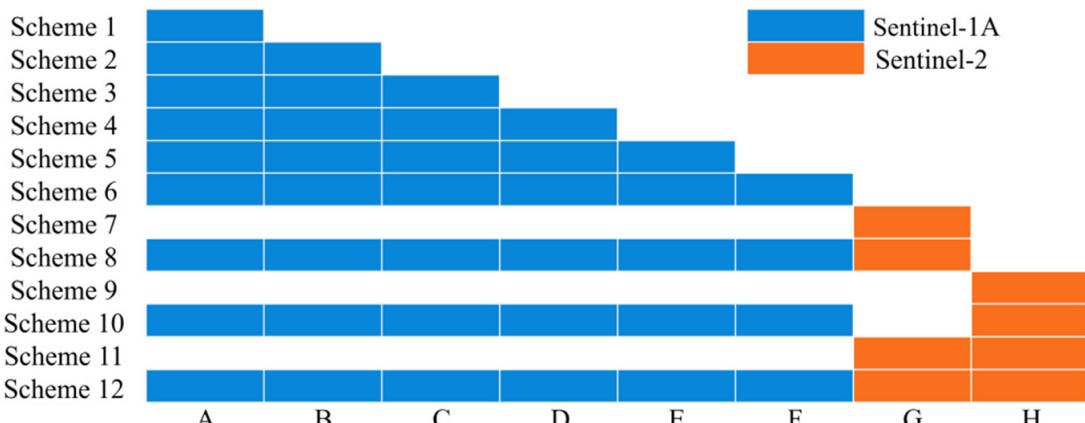

**Figure 6.** Experimental schemes A, B, C, D, E, and F represent the SAR images of sowing, seedling and tillering, tillering and over-wintering, over-wintering and reviving, jointing and heading, and flowering and maturing winter wheat, respectively; G and H represent the optical images of before-wintering and after-wintering winter wheat, respectively.

### 3.4. Random Forest Algorithm

The random forest (RF) algorithm was used as the classification method in this study, which is an integrated learning classifier composed of several decision trees and a voting mechanism [43,44]. Compared with other machine learning algorithms, the RF algorithm has the advantages of good robustness, fast classification speed, high classification accuracy, and not being easy to overfit. It has been widely used in the field of agriculture. Usually, the sample data are divided into a training dataset for model construction and a test dataset

for model verification. When constructing an RF model, two basic parameters must be defined: the number of decision trees and the number of feature variables.

### 3.5. Accuracy Assessment

In this study, the confusion matrix calculated based on the real ground reference data was used to quantitatively evaluate the accuracy of the classification results. Then, the classification results were evaluated by several indicators that are calculated by the confusion matrix, including overall accuracy (OA), kappa coefficient (Kappa), mapping accuracy (MA), user's accuracy (UA), and F1 measure ($F_1$) [17,45]. OA and Kappa are overall metrics to evaluate the classification results; MA, UA, and $F_1$ are indicators to measure the accuracy of feature classification, and they can reflect the quality of feature classification from different aspects.

### 3.6. Assessment of Feature Variable Importance

The feature variable importance score reflects the relative importance of each feature variable in the prediction process [17,26]. When constructing the RF algorithm, the bootstrap sampling technique was used to obtain the training subsets from the original training dataset and the unselected data from the out-of-bag data (OBB). OBB can validate the weights of the input feature variables and express the weight score of each feature variable in terms of the average precision reduction value. This study used the explain function of the GEE platform to obtain the weight of each feature variable and calculate the importance score of the feature bands.

## 4. Results

### 4.1. Accuracy of Experimental Schemes

The extraction accuracy of different schemes is shown in Figure 7. The results show that: (1) When only Sentinel-1A images were used, the extraction accuracy was improved with the aggregation of the images in the growth period. The extraction accuracy on SAR images in the whole growth period can basically meet the mapping requirements of winter wheat (OA and Kappa were 80.1% and 0.733, respectively). (2) When only Sentinel-2 images were used, the extraction accuracy on the images of after-wintering winter wheat was much higher than that on the images of before-wintering winter wheat, and the OA on after-wintering winter wheat images was 86.7%. After the integration of before-wintering and after-wintering images, the extraction accuracy was improved by different degrees. This is consistent with the result that the winter wheat acreage was extracted by integrating the SAR images of multiple growth periods. (3) By using the integrated Sentinel images for classification, the extraction accuracy was improved compared with that obtained by using the images of a single sensor type. After the integration of all images, the OA and Kappa were 92.7% and 0.902, respectively. The OA was 12.6% higher than that obtained by using the SAR images in the whole growth period, indicating that a more accurate spatial distribution map of winter wheat was obtained.

### 4.2. Mapping Results of Winter Wheat in Henan Province

After the integration of Sentinel-1A images in the whole growth period and Sentinel-2 images before-wintering and after-wintering (scheme 12), the spatial distribution map of winter wheat in Henan Province with a spatial resolution of 10 m was obtained by using the RF algorithm. Figure 8a indicates that winter wheat is mainly distributed in the central-eastern plain and Nanyang basin of Henan Province, showing the characteristics of concentrated continuous distribution and partially fragmented cultivation. This result is basically consistent with the findings of previous studies [28,29]. Figure 8 enlarges some details, and the detailed images (Figure 8c) are compared with the reference images (Figure 8b). It can be seen that most of the winter wheat acreage has been correctly classified.

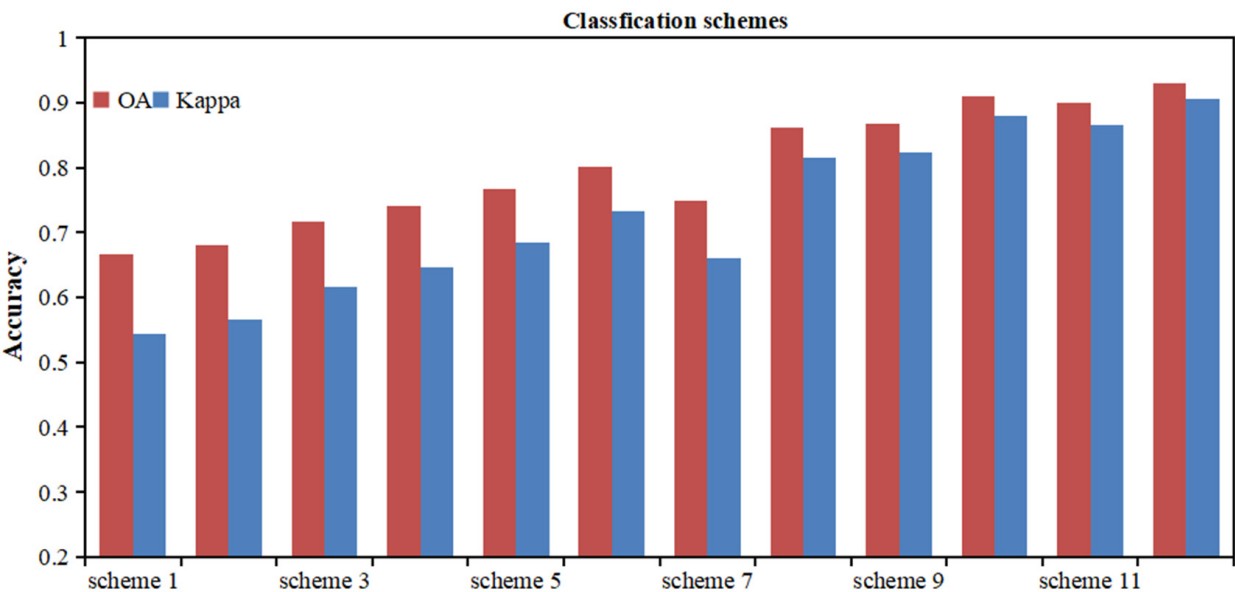

**Figure 7.** Overall accuracy (OA) and kappa coefficient (Kappa) of different classification schemes.

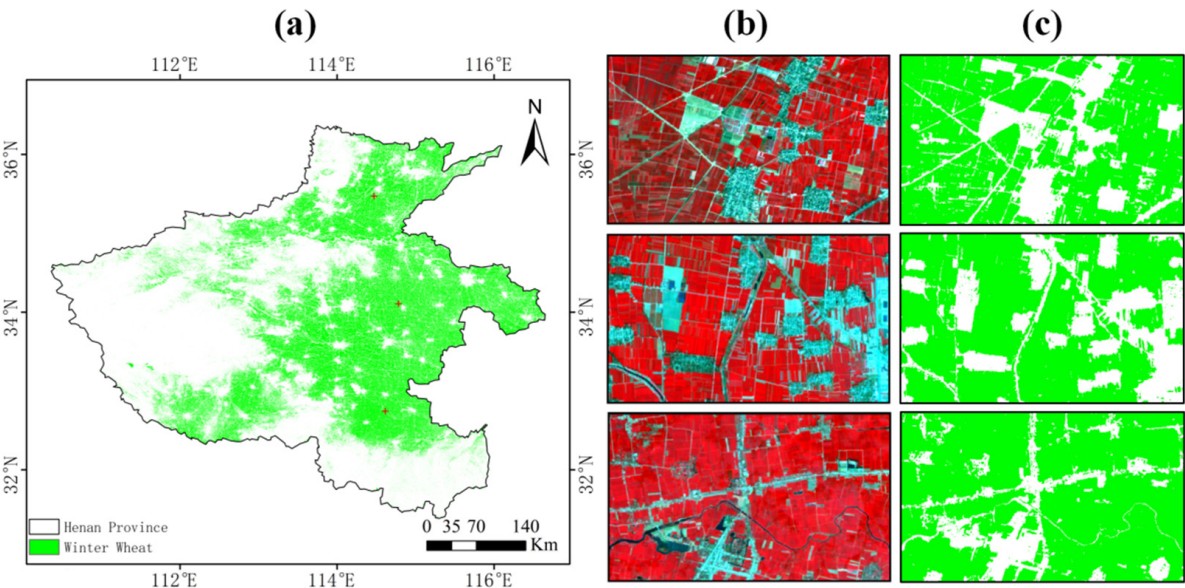

**Figure 8.** The spatial distribution map of winter wheat in Henan Province in 2018–2019 cropping season (**a**). The Sentinel-2 false-color images (NIR/ Red/Green) (**b**) and the zoomed-in distribution map of the winter wheat in three example locations (**c**) from Henan Province.

As shown in Table 4, the MA was 95.0%, and the $F_1$ was 0.941. Meanwhile, the area of winter wheat was calculated according to the extraction results. The area of winter wheat in Henan Province is 56,075 km$^2$, which is 726 km$^2$ less than the area estimated by the official statistics of China in 2019 (57,066 km$^2$). The extraction of winter wheat has high accuracy and is relatively stable. Winter wheat can be well distinguished from water, buildings, and others. However, vegetation may be misclassified into winter wheat, and this is caused by the similarity of vegetation to winter wheat in terms of characteristic variables. The results are informative for the development of agricultural policies.

**Table 4.** Confusion matrix of the winter wheat distribution map in Henan Province.

| Land-Cover Types | Classification Results | | | | | | | |
|---|---|---|---|---|---|---|---|---|
| | **Winter Wheat** | **Vegetation** | **Buildings** | **Water** | **Others** | **Sum** | **MA** | **F₁** |
| Winter wheat | 246 | 10 | 2 | 0 | 1 | 259 | 95.0% | 0.941 |
| Vegetation | 18 | 224 | 1 | 1 | 3 | 248 | 90.7% | 0.905 |
| Buildings | 0 | 5 | 132 | 1 | 3 | 141 | 93.6% | 0.939 |
| Water | 0 | 2 | 0 | 65 | 0 | 67 | 97.0% | 0.970 |
| Others | 0 | 7 | 5 | 0 | 79 | 91 | 86.8% | 0.893 |
| Sum | 264 | 248 | 140 | 67 | 86 | | OA = | Kappa |
| UA | 93.2% | 90.3% | 94.3% | 97.0% | 91.9% | | 92.7% | = 0.902 |

### 4.3. Comparison of Spatial Details and Quantitative Evaluation

According to the study purpose, the extraction results of experimental schemes 6, 11, and 12 are displayed (Figure 9). This study selected three regions and drew the classification details. The Sentinel image with a spatial resolution of 10 m had a poor extraction effect on narrow linear features. These three extraction schemes tended to mistakenly classify roads as winter wheat. In contrast, the SAR image (scheme 6) had a better extraction effect on roads, but it may misclassify vegetation to winter wheat, resulting in the overestimation of winter wheat acreage. The results show that the extraction effect of the integration of Sentinel images (scheme 12) is better than that of using the SAR images in the whole growth period (scheme 6) and optical images (scheme 11). Though the use of SAR images in the whole growth period can effectively extract a wide range of winter wheat planting areas, there is still "pepper and salt noise", and this problem can be reduced by image integration.

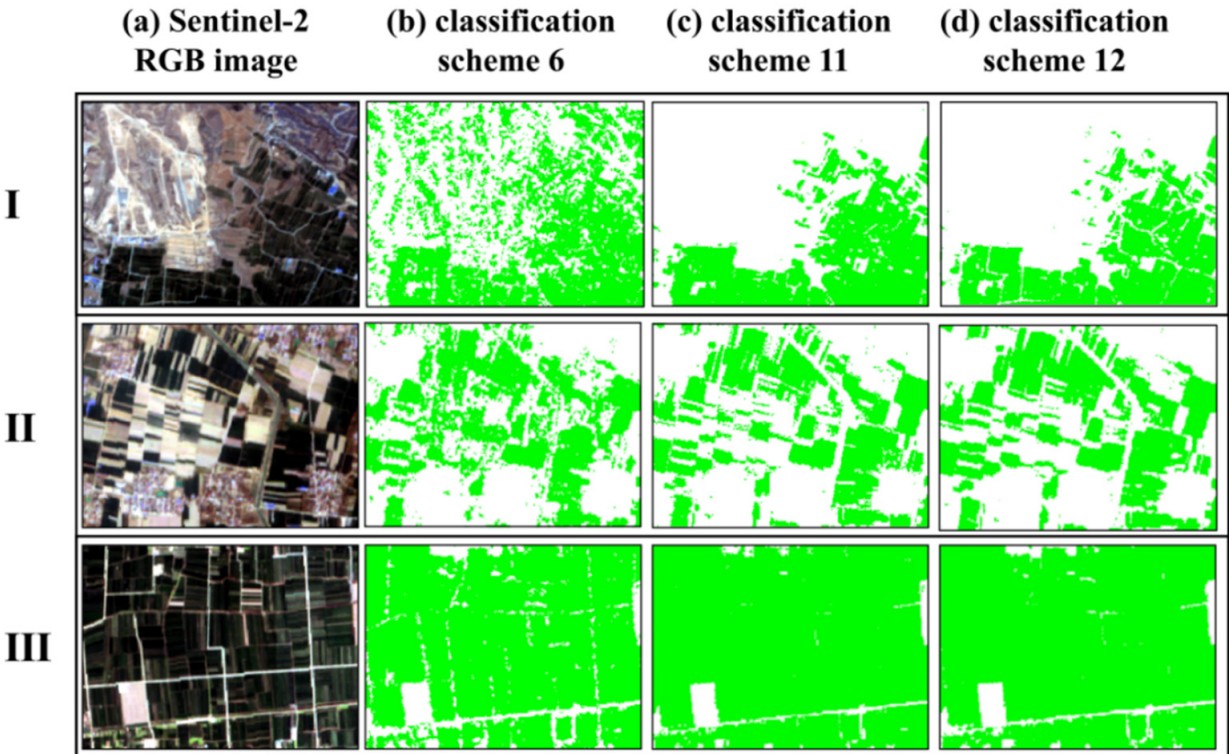

**Figure 9.** Comparison of spatial details of four example locations, location I (114.070° N, 35.576° E), II (114.910° N, 33.790° E), III (114.862° N, 32.585° E). (**a**) Sentinel-2 RGB composite images. The winter wheat maps derived using the (**b**) classification scheme F, (**c**) classification scheme K, (**d**) classification scheme L.

*4.4. Feature Variables Importance*

As can be seen from Figure 10 (left), when only Sentinel-1A images were used to extract winter wheat acreage, the importance scores of SAR images of overwintering and reviving, jointing and heading, and flowering and maturing were high, all about 0.18, which played a dominant role in the classification results. Compared with polarization features, the importance score of texture features is relatively low. After the integration of Sentinel-2 images, the influence of the texture features on the classification results was low (Figure 9, right). This is related to the calculation of texture features from polarization features. When using integrated images to extract winter wheat acreage, the importance scores of spectral features and polarization features were higher than those of texture features. The feature variable with the highest score was the spectral feature, which indicates the importance of optical images in extracting winter wheat acreage.

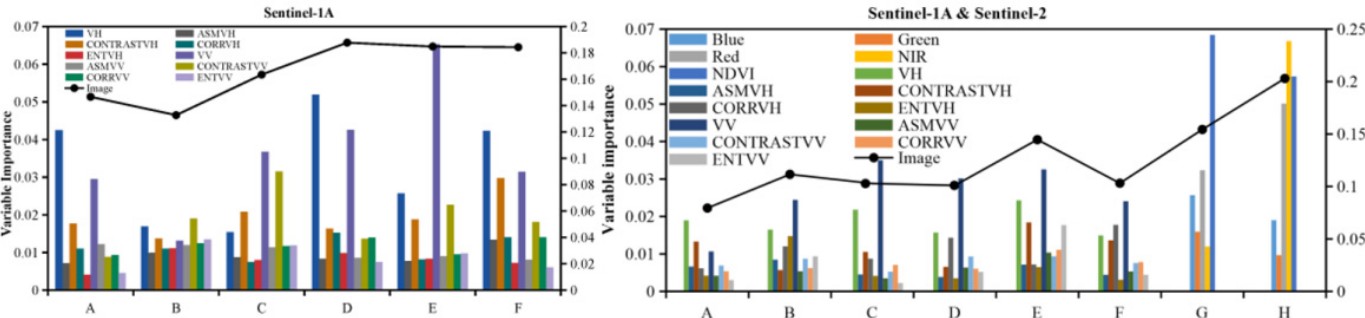

**Figure 10.** Importance of feature variables for RF classification on different data. The results showed the importance scores of classification scheme 11 (**left**) and classification scheme 12 (**right**).

**5. Discussion**

*5.1. Image Aggregation Method*

For large-scale remote sensing mapping of winter wheat, selecting a reasonable image aggregation method can improve the mapping accuracy and efficiency. A large number of high-quality images in the study area provide a data basis for the implementation of the aggregation scheme (Figure 2). Based on the imaging principle of the sensor, this study designed image aggregation methods, respectively.

Due to the influence of cloudy and rainy weather, there may be no optical image coverage in a certain growth period of winter wheat. The complete image is generally aggregated by increasing the image aggregation time window. However, a larger image aggregation time window will span more growth periods. The aggregated images are not representative and cannot accurately reflect the phonological features of winter wheat. The NDVI curve of winter wheat has the characteristic of "two peaks and one valley", which is different from that of other land-cover types [46]. Before overwintering, winter wheat is sparse and easily influenced by soil background; after overwintering, winter wheat grows rapidly, and the variation trend of NDVI is similar to that of vegetation, which is easy to be affected by vegetation. Besides, the NDVI curves of winter wheat before and after overwintering both show an increasing trend [16]. The overwintering of winter wheat maintains for two months, but the satellite images in this period are affected by rain and snow. It is difficult to aggregate a complete phase of optical images. Moreover, the NDVI characteristics of winter wheat do not change significantly during the overwintering period.

Compared to optical images, SAR images have a robust data source in all weather conditions. The Sentinel-1A satellite can provide one SAR image for 12 days. In the early period of winter wheat sowing, the plants are low and sparse. Meanwhile, the influence of soil background is dominant, and the backscattering coefficient increases. When winter wheat begins to develop, the stem density increases gradually [47,48]. At this time, the influence of rainfall causes the curve to fluctuate upward, but the backscattering coefficient decreases as a whole. After jointing, winter wheat gradually begins to senescence, and

the water content in the plant decreases. Meanwhile, the influence of soil background and the backscattering coefficient both increase. The variation trend of the backscattering coefficient can be fully captured by image aggregation with the growth period of winter wheat as the time window. This method makes full use of the phenological characteristics of winter wheat, and it solves the problem of extracting winter wheat planting area from SAR median images.

## 5.2. Potential of Using SAR Images of the Full Growth Period to Extract Winter Wheat Acreage

Our study shows that it is feasible to use SAR images of the whole growth period to draw the spatial distribution map of winter wheat at the provincial level. The radar imaging technology can acquire information on crop canopy structure and plant water content [49,50]. The extraction accuracy increases with the accumulation of growth periods, and this is related to the fact that the image integration of multi-growth periods can reduce speckle noise. The integration of SAR images of five growth periods contributes to an extraction accuracy close to 80%, which helps to understand the spatial distribution of winter wheat and provides reliable information for the ministry of agriculture to evaluate food security. In Figure 8, there is a lot of "salt and pepper noise" in the classification results based on SAR images in the full growth period, and the interior of the plot is relatively rough. Also, the classification boundary between winter wheat and other land-cover types is fuzzy. Though there are wrong classifications and missing classifications, most pixels are correctly classified. For the planting areas with large distribution and concentrated contiguity of winter wheat, the SAR image of the full growth period can achieve better classification results. For the winter wheat planting areas with complex and scattered ground features, there are great defects, and the extraction accuracy needs to be improved [24,51,52].

When using the SAR image of the full growth period for classification, polarization feature and contrast texture feature are the two most important features in the importance score of feature variables. The inclusion of texture features can improve the extraction accuracy, but the importance scores of texture features are relatively low. Since texture features are calculated based on polarization features, they have a high correlation with polarization features. Therefore, the contribution of texture features to extraction results is relatively low. Compared with other texture features, contrast features contain more information in space [26,44]. The SAR images of overwintering and reviving, and flowering and maturing contribute greatly to the extraction results of winter wheat. During the period of overwintering and reviving, winter wheat grows vigorously, and the plant height increases continuously, which results in a low backscattering coefficient. At the flowering period, the water content of the winter wheat plant decreases continuously, and the backscattering coefficient is increased. In the later growth period, the polarization features of winter wheat change obviously. This helps to distinguish winter wheat from other land-cover types, and it is conducive to the extraction of winter wheat.

## 5.3. Advantages of Image Integration for Extraction of Winter Wheat

This study demonstrated that image integration is more advantageous for mapping the spatial distribution of winter wheat. The successful launch of the radar imaging satellite brings the development of crop extraction technology into a new stage [53,54]. For different imaging methods, the data sources complement each other and play their respective advantages in crop extraction. The multi-temporal remote sensing images could improve crop classification accuracy. However, the phenological characteristics vary widely in different growth periods. It is crucial to choose the growth period reasonably. Different from the previous studies, this study integrated remote sensing images with the growth period of winter wheat as the time window. The Sentinel-1A images of the full growth period and Sentinel-2 images before and after overwintering were used to map the spatial distribution of winter wheat, and the experimental results were reliable.

Previous studies have shown that vegetation index features are sensitive to vegetation and have strong advantages in vegetation identification [55,56]. This was reflected in the feature variable importance analysis. The importance scores of optical image bands are higher than those of SAR image bands. The optical image after overwintering has a great impact on the extraction accuracy, and this is due to the fact that winter wheat after overwintering is easy to distinguish from other ground object types. However, in the case of less optical image phase data, the subdivision of vegetation types will produce a large number of misclassifications and missing classifications. SAR images are sensitive to crop structure and can reflect the effects of crop planting density, plant moisture content, and soil background [57]. The image integration can make up for the shortcomings of SAR image and optical image, and it is very effective to improve the remote sensing identification accuracy of winter wheat. The use of integrated Sentinel images contributed to higher extraction accuracy than the use of sensor images alone, and the OA and Kappa were 92.7% and 0.902, respectively. Meanwhile, the extraction results of integrated images also have less "pepper and salt noise". Therefore, SAR images can be used as an auxiliary data source to help optical images for winter wheat acreage extraction [58]. Our method provides a way to determine the early identification time of winter wheat.

### 5.4. Limitations and Prospects

GEE remote sensing cloud platform makes remote sensing mapping at a large scale possible, and the powerful data analysis capability enables the implementation of the experiment in this study. Based on the GEE platform, this study extracted the 2018 winter wheat acreage at the provincial scale using integrated Sentinel images and mapped the spatial distribution of winter wheat. The GEE platform has many years of remote sensing data sources, and the full use of these images allows the production of winter wheat time series products [59]. The use of time-series products to monitor the spatial and temporal variation of winter wheat can provide reliable information for governments to make plans [16]. Our future study will focus on mapping the spatial distribution of winter wheat over longer time series.

Our spatial distribution map of winter wheat with 10 m spatial resolution has high extraction accuracy and reliable results. The integration of optical and SAR images can reflect crop phenological characteristics from different aspects. The integration of images from different sensors can take full use of the information of optical images and give full play to the advantages of SAR images. This study combined the phenological characteristics of winter wheat with the image characteristics and integrate the images separately. This can provide an effective reference for winter wheat mapping in other regions. During the winter wheat observation period, a small amount of rape and garlic will be planted in Henan Province, which will still cause some disturbance to the results. This study did not consider the interference caused by rapeseed and garlic. In our future work, the influence of zoning on crop planting information extraction will be investigated.

## 6. Conclusions

In this study, the Sentinel-1A images of the full growth period and Sentinel-2 images before-overwintering and after-overwintering were used as data sources to map the spatial distribution of winter wheat in Henan Province in 2018 based on the RF algorithm. The feasibility and potential of using SAR images in the full growth period and integrating images to extract winter wheat acreage were explored. Our study determined the following. (1) The aggregation of Sentinel-1A images based on the winter wheat phenological period can provide sufficient data sources for experiments and facilitate the construction of phenological indicators. The aggregation of Sentinel-2 images based on the MODIS-NDVI curve can effectively take full use of the optical image and enhance the image information. (2) The use of SAR images in the full growth period under the combination of polarization features and texture features can achieve winter wheat acreage extraction. The extraction effect varies for winter wheat growing areas with different plot types. The analysis of the

importance scores of feature variables indicated that polarization features and contrast texture features dominated the extraction results. (3) The use of integrated Sentinel images achieved higher extraction accuracy than the use of sensor images alone, and the identification effect was better. In the case of insufficient optical images, integrated images can be efficient for monitoring winter wheat. The method proposed in this paper has a great potential to be extended to remote sensing mapping in larger study areas with more complex crop cultivations.

**Author Contributions:** Conceptualization, C.L.; methodology, W.C.; software, W.C.; formal analysis, Y.L.; writing original draft preparation, W.C.; writing review and editing, C.L.; validation, W.C.; investigation, Y.W. (Yilin Wang) and W.Z.; supervision, Y.W. (Yu Wang); visualization, J.L.; funding acquisition, C.L. and C.M. All authors have read and agreed to the published version of the manuscript.

**Funding:** This study was supported by the Natural Science Foundation of China (41871333), the Scientific and Technological Innovation Team of Universities in Henan Province (22IRTSTHN008), the Important Project of Science and Technology of the Henan Province (212102110238) and Key scientific research project of Henan college and university (20B420002).

**Institutional Review Board Statement:** Not applicable.

**Informed Consent Statement:** Not applicable.

**Data Availability Statement:** MODIS, Sentinel-1 and Sentinel-2A/B data are openly available via the Google Earth Engine.

**Acknowledgments:** We are grateful to the anonymous reviewers whose constructive suggestions have improved the quality of this study. We wish to express our gratitude to USGS and GEE platform for supplying MODIS and Sentinel data.

**Conflicts of Interest:** The authors declare no conflict of interest.

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
