# Peer review of "Mapping Winter Wheat with Optical and SAR Images Based on Google Earth Engine in Henan Province, China"

_remotesensing, doi:10.3390/rs14020284_

Round 1

Reviewer 1 Report

This study investigated the integration between Sentinel-1A and Sentinel-2 to map the acreage of winter wheat in Henan province. The results implied that the integration of the SAR imagery acquired during the entire growth season obtained better results than a single acquisition. The combination between microwave and optical imagery improved the classification accuracy for the winter wheat. This paper collected and analyzed a large number of data, and tested the feasibility of the Random Forest algorithm in delineating the winter wheat. My major concern is the innovation of this paper, as the Random Forest is a well-developed algorithm in image classification and retrievals.

  • The revisit cycle of Sentinel-1A is 12 days. In Table 3, how did the authors acquire a large number of images during the growth season of winter wheat?
  • Figure 3: It is better to add the data marker in the figure so that we can be informed where are the Sentinel-1A acquisition? Why did the authors use the Bessel curve to smoothen the backscattering curves in Figure 3?
  • I cannot understand why the authors selected the Sentinel-1A, rather than both Sentinel-1A and 1B to shorten the satellite revisit interval.
  • Figure 2: in the flowchart, please correct the error in the top-left box into “Sentinel-2 image”.
  • Section 3: It is not clear how the Sentinel-1A and Sentinel-2 are integrated to map the winter wheat. The integration approach is not clear in the current version. 
  • Eq. (1): I do not think it is necessary to show the formula of NDVI, as this is well known in the community.

Reviewer 2 Report

This study investigated the feasibility of extracting the spatial distribution map of winter wheat in Henan Province by using synthetic aperture radar (SAR, Sentinel-1A) and optical (Sentinel-2) images.

I find the manuscript to be very interestingly. I only have some comments to be address as followings:

Introduction

This section needs to be concise

L41-42 The expression of spatial distribution is contradictory

L54-61 The summary of references needs to be concise. At the same time, other researchers have done a series of crop mapping work in Henan Province Based on sentinel and Landsat images, e.g. mapping winter crops, cropping intensity. Please refer to Pan, L., Xia, H., Zhao, X., Guo, Y., Qin, Y., 2021. Mapping Winter Crops Using a Phenology Algorithm, Time-Series Sentinel-2 and Landsat-7/8 Images, and Google Earth Engine. Remote Sensing 13, 2510. and Mapping winter crops in China with multi-source satellite imagery and phenology-based algorithm. Please compare and discuss the results of this paper with above articles.

L62-69, L70-86 the introduction of GEE needs to be concise

Study area and datasets

L23-24 in the south of Hean.

L125 REVISED TO from north to south is 10.5-16.7℃

L124-L126 the value of temperature, sunshine, precipitation based which data, years, please list or give reference.

L152, THE SELECTED PERIODS IS inappropriate (The selected study period is inappropriate), The sowing and harvesting dates of the north, South, East and west of Henan Province are not consistent, , which may exceed this periods.

2.2.2 Sampling strategy needs to be improved,

L245  the Since most of the vegetation in Henan Province is deciduous vegetation, What is the basis of this statement. the paper of “Mapping Winter Crops Using a Phenology Algorithm, Time-Series Sentinel-2 and Landsat-7/8 Images, and Google Earth Engine” give the evergreen forest, please include

Figure 5 different region have different phenology, how do control the proper date for different combined schemes.

Results

In Section “4.2 Mapping results of winter wheat in Henan Province” please give area of winter wheat in Henan PROVINCE AND COPARE WITH THE SOME STUDY AREA IN PAPER,Mapping Winter Crops Using a Phenology Algorithm, Time-Series Sentinel-2 and Landsat-7/8 Images, and Google Earth Engine

Discussion

5.Discussion, the mapping results need compare with results of the same study area, and compare the mapping method combine multi-remote sensing data

Reviewer 3 Report

In the article “Mapping Winter Wheat with Optical and SAR Images Based on  Google Earth Engine in Henan Province, China”  an important  issue is considered. A lot of work has been done by the authors of the article.

While reading the manuscript, I had a question about the advisability of using data with low spatial resolution (MODIS) in areas with narrow and small fields. MODIS has one advantage - good periodicity, but a big disadvantage - low spatial resolution. When processed, such data will contain mixed pixels with different classes of the earth's surface, different crops. After carefully reading the manuscript, I would suggest making certain changes.

Detailed recommendations are provided below.

85 – indicate the authors of the harvest monitoring manuscripts based on SAR data

142 –  a  more detailed processing steps should be provided

150 – what cloud masking method was used and why

180-189 “Methodology” – a  more detailed description of the method is needed.

  1. “3.4. Calculation of feature variables” – justification is needed why window 3x3 was chosen

Round 2

Reviewer 2 Report

ALL comments have been address, and accept in present form.